# Learning to Configure Computer Networks with Neural Algorithmic Reasoning

**Luca Beurer-Kellner**[1, *]  **Martin Vechev**[1]  **Laurent Vanbever**[1]  **Petar Veličković**[2]

[1]ETH Zurich, Switzerland  [2]DeepMind

https://github.com/eth-sri/learning-to-configure-networks

## Abstract

We present a new method for scaling automatic configuration of computer networks. The key idea is to relax the computationally hard search problem of finding a configuration that satisfies a given specification into an approximate objective amenable to learning-based techniques. Based on this idea, we train a neural algorithmic model which learns to generate configurations likely to (fully or partially) satisfy a given specification under existing routing protocols. By relaxing the rigid satisfaction guarantees, our approach (i) enables greater flexibility: it is protocol-agnostic, enables cross-protocol reasoning, and does not depend on hardcoded rules; and (ii) finds configurations for much larger computer networks than previously possible. Our learned synthesizer is up to $490\times$ faster than state-of-the-art SMT-based methods, while producing configurations which on average satisfy more than 92% of the provided requirements.

## 1 Introduction

Configuring large-scale networks is a challenging and important task as network configuration mistakes regularly lead to massive internet-wide outages affecting millions (resp. billions[2]) of Internet users [35, 25]. Typically, network operators provide a router-level configuration $W$ which, after applying protocols such as shortest-path routing, induces a certain forwarding behaviour FWD as illustrated in Figure 1. As this remains a challenging task, much recent research has focused on automating configuration by leveraging synthesis techniques [15, 5, 31]: A synthesizer is used to automatically generate a router-level configuration $W$ that, after applying routing protocols results in forwarding behavior that satisfies a given specification $S$ on how traffic should be routed.

**SMT-based Synthesis**  Due to the hardness of the configuration synthesis problem [7], many effective tools in this domain [15, 14] resort to satisfiability modulo theory (SMT) solvers, which employ search-based procedures to find a solution to a set of first-order logic constraints. This enables comprehensive and exact synthesis by modelling network behavior in first-order logic. However, these tools are typically protocol-specific, hand-coded, and can exhibit discrepancies in behavior when compared to actual router hardware [6]. Most importantly, however, they can be very slow or fail to complete for large networks. For example, the state-of-the-art SMT-based tool NetComplete [15] requires more than 6 hours to synthesize a configuration for a network with 64 nodes, for which other SMT-based tools like SyNET [14] take even longer (> 24 hours) [15]. Non-SMT-based tools such as Propane [4] or Zeppelin [39] have achieved better performance, but at the cost of generality.

---

*Correspondence to `luca.beurer-kellner@inf.ethz.ch`.

[2]As of 2021, Facebook reportedly has 2.9 billion monthly active users [1].

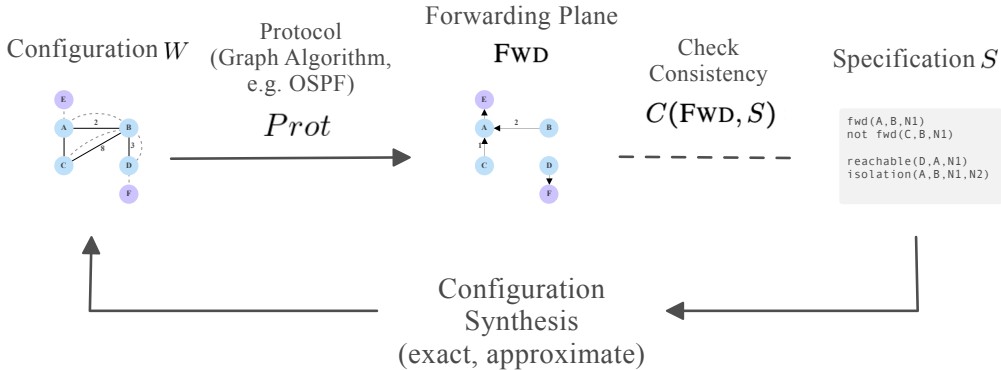

Figure 1: **Network configuration synthesis.** The goal of configuration synthesis is to find a configuration $W$ that maximizes consistency $C(\text{FWD}, S)$ for a given specification $S$.

**Addressing the scalability barrier**   The reason for the wide-spread use of SMT in configuration synthesis is the inherent computational complexity of the underlying synthesis problem, parts of which have been shown to be NP-hard [7, 16, 44]. This means that any exact synthesis method is bound to run into scalability issues (as do all SMT-based methods). However, to be practically useful, a synthesizer must scale to the size of real-world networks and the frequency at which configurations are updated. For example, in a Tier-1 ISP, network operators modify their configurations up to 20 times per day, on average [36]. We argue that one way to address this scalability barrier, is relaxing the configuration synthesis problem to admit approximate solutions with high utility – configurations which may not always satisfy all requirements of a given specification but may satisfy almost all of them. Such a configuration would be a much better starting point for a network operator than having no automated support whatsoever. The core technical challenge then is coming up with a strategy likely to find solutions with high utility.

**This Work: enabling fast and scalable configuration via neural algorithmic reasoning**   We address this challenge and present the first learning-based framework for approximate configuration synthesis. Our relaxed formulation allows us, for the first time, to apply end-to-end learning to the problem of network configuration synthesis, thereby enabling fast and scalable configuration synthesis with almost interactive response times (<90s even for large networks). Technically, we leverage the observation that routing protocols can often be formulated as Bellman-Ford style graph algorithms, a class of problems that has recently been studied in the area of neural algorithmic reasoning (NAR) [43]. Connecting the two fields and building on ideas from NAR, we are able to train a graph-based neural model with a strong inductive bias to learn an inverse mapping from specifications back to network configurations: Our model learns how to perform synthesis from a dataset of (specification, configuration) pairs obtained by simulating the involved protocols and observing the computed forwarding state. With this method, we can support cross-protocol reasoning and do not have to manually provide any hardcoded synthesis rules. Concretely, we introduce a generic embedding scheme for topologies and configurations, making our method protocol-agnostic. During synthesis, given a specification, our model predicts distributions of network configurations from which we can sample possible results.

**Main Contributions**   Our core contributions are:

- We formulate a relaxation of the exact configuration synthesis problem, which enables fast and scalable network configuration, amenable to learning-based techniques (Section 2).

- We propose a NAR-based neural network architecture for learning synthesizer models that rely on a graph-based encoding of topologies and configurations, and a strong inductive bias towards an iterative synthesis procedure (Section 4).

- We conduct an extensive evaluation of our learning-based synthesizer with respect to both precision and scalability. We demonstrate that our learned synthesizer is up to $490\times$ faster than a state-of-the-art SMT-based tool while producing high utility configurations which on average satisfy $> 93\%$ of provided constraints (Section 5).

## 2   Configuration Synthesis: Exact and Learned

We first state the general configuration synthesis problem and explain why it is hard to solve. We then present a rather different approach based on learning that addresses the scalability barrier of traditional synthesis.

**Forwarding Behavior and Specifications**   We focus on the level of the *forwarding plane* of a network. This means we consider how a network forwards traffic, given a packet with a certain destination. The forwarding plane is determined by a distributed computation that depends on the different routing protocols in use. More formally, we define the forwarding plane FWD as follows:

$$\text{FWD} := Prot(W; T)$$

$Prot(W; T)$ corresponds to the result of applying routing protocols to the network topology $T$ and the configuration $W$ (e.g. link weights). In the following, we omit $T$ as it remains fixed in synthesis. The resulting forwarding plane FWD can be understood as a directed graph superimposed on topology $T$. It specifies a subset of links that are used to forward packets. To illustrate consider Figure 1: applying the routing protocols yields forwarding plane FWD which corresponds to the subset of links $(C, A), (B, A), (A, E), (D, F)$. The other links of the network are not part of the forwarding plane and will thus not be used to forward traffic.

Given FWD , we consider a forwarding specification $S := \{R_i\}_i$ as an input to the synthesis problem. Each requirement in $S$ is modelled as a function $R_i$, where $R_i(\text{FWD}) = 1$ if FWD satisfies the requirement and $0$ otherwise. Practical example requirements include reachability, traffic isolation or specifying concrete forwarding paths.

**Exact Configuration Synthesis**   We formulate the general configuration synthesis problem as the following optimization objective:

$$W^\star := \underset{W \in P(W)}{\arg\max} \; C(Prot(W), S) \quad \text{where} \quad C(\text{FWD}, S) := \frac{\sum_{R_i \in S} R_i(\text{FWD})}{|S|} \tag{1}$$

$P(W)$ denotes the set of all possible configurations and $C(\text{FWD}, S)$ is the *specification consistency* of a forwarding plane FWD w.r.t specification $S$. In traditional, exact configuration synthesis, this objective is solved by limiting the search to globally-optimal configurations such that $C(\text{FWD}, S) = 1.0$, i.e. *all* requirements must be satisfied. Such exact methods typically resort to SMT solvers because of the hardness of the underlying problem: configurations comprise a large number of tunable parameters, where the execution of several interacting protocols yields the overall forwarding state. Even worse, parts of the configuration synthesis problem have been shown to be NP-hard [7, 16, 44]: For example, already the subproblem of finding link weights that yield a given set of forwarding paths under shortest-path routing is NP-hard [7]. This makes scaling exact synthesis to real-world networks extremely challenging.

**Learning-Based Synthesis**   To enable fast and scalable configuration synthesis, we propose to relax both the optimality as well as the rigid satisfaction requirements w.r.t the specification $S$. Concretely, we relax the set of admissible solutions to include configurations that are not optimal, but still satisfy a large number of provided requirements. Note that this is not the same as merely allowing solutions with $C(\text{FWD}, S) < 1.0$, because maximum satisfiability does not relax the hardness of the problem. Instead, we propose to search for near-optimal, good solutions and rely on the value of specification consistency $C$ as a measure of quality.

An approximate synthesis formulation relaxes the hardness of the problem, however, it also leads to the difficult technical challenge of finding solutions with high utility (e.g., where many requirements are satisfied). To address this challenge, we propose a rather different approach where we learn synthesis from data. Concretely, we learn an inverse mapping that attempts to predict approximate solutions $\hat{W}^\star$ with high utility, as guided by the following objective:

$$\hat{W}^\star = Prot^{-1}(\text{FWD}) \;\; \text{s.t.} \;\; C(\text{FWD}, S) \text{ is high}$$

This can be implemented as a synthesizer model $M_{Syn}$ which produces a solution given just the topology $T$ and the specification $S$:

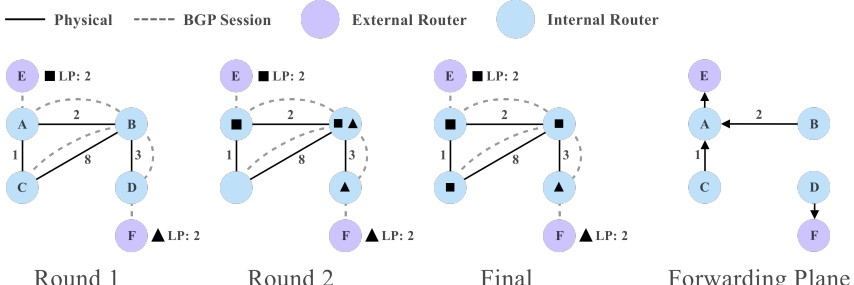

Figure 2: An illustration of propagating BGP route announcements in a small network, including internal peers (blue), external peers (purple), BGP sessions as well as physical links. The last graph illustrates the resulting forwarding plane after applying BGP/OSPF.

$$\hat{W}^{\star} = M_{Syn}(S;T)$$

We propose a learning-based approach for training such synthesizer models based on neural algorithmic reasoning (cf. Section 4). First, however, we discuss the routing protocols that make up function $Prot$ and how they relate to graph algorithms and by extension to NAR.

## 3 Routing Protocols as Graph Algorithms

In practice, the routing protocols defining $Prot$ are implemented as distributed systems in which multiple peers communicate to determine the network's forwarding state. However, theoretical work on routing algebras [21] has shown that the underlying computation can be understood as a traditional message-passing graph algorithm. As a consequence, many routing protocols can be formulated as Bellman-Ford (BF) style propagation processes. This class of problems has also recently been subject to work on NAR [43, 41] and algorithmic alignment [45]. The authors of these works demonstrate that neural networks are capable of closely imitating BF-style algorithms when provided with a suitable inductive bias. Based on this insight, NAR proposes to replace traditional algorithms with neural networks to learn improved algorithmic procedures or extend existing algorithms to be applicable to raw data [41]. Following the idea of NAR, we implement a synthesizer model for network configurations as an iterative Graph Neural Network (GNN) to learn $Prot^{-1}$ by relying on a Bellman-Ford style inductive bias.

**Synthesis Setting**   Our approach is general for the domain of networks, but we focus on two widely-used routing protocols: (1) Open Shortest Path First (OSPF) [30] – it uses link weights to route traffic along the shortest path towards the destination, and the (2) Border Gateway Protocol (BGP) [32], used to exchange reachability information, mostly on the level of larger backbone networks. When using BGP, a routing destination announces its existence to other networks and routers by sending out BGP announcements. Receivers of announcements then choose to pass them on to other peers, redistribute them internally and/or modify them according to a set of decision rules. BGP and OSPF interact, for instance, BGP will consider the OSPF cost of internal destinations in its routing decisions. This means, that effective BGP/OSPF synthesis tools must implement cross-protocol reasoning, configuring BGP and OSPF, such that together they yield the desired forwarding behaviour.

**Example**   To provide a basic intuition about BGP/OSPF routing, consider the simple example network in Figure 2. We apply the two protocols to obtain the forwarding plane. Physical OSPF edges are labelled with a corresponding link weight determining the OSPF cost of paths through the network. Dotted lines represent designated BGP edges, used to propagate BGP announcements. Our network imports multiple BGP announcements ■ and ▲ from E and F respectively. Both represent a route to the same destination. As shown, the announcements have a so-called BGP local preference of 2 (we ignore other BGP properties in this example). The announcements are propagated through the network and the best route is selected according to a designated BGP decision procedure. Figure 2 shows the intermediate states of this propagation process. As both announcements have the same local preference value, the decision cannot eliminate based on that. Instead, in round 2, node B

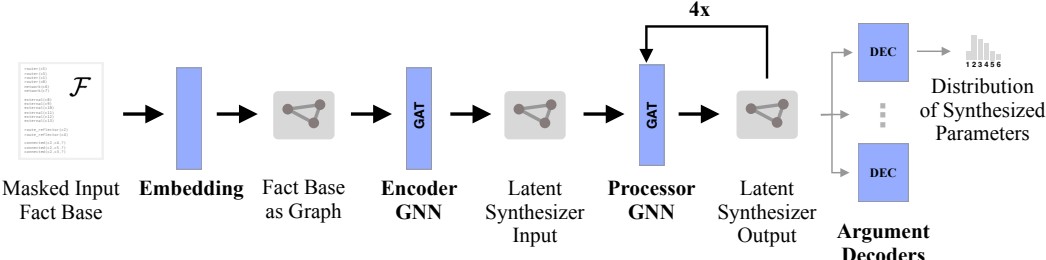

Figure 3: **Neural synthesizer architecture**. A provided input fact base is first embedded as a graph including the masked to-be-synthesized parameters. Then encoder GNN, processor GNN and decoder networks are applied to obtain a distribution over synthesized parameters.

selects ■ over ▲ due to a lower OSPF cost (shorter path) of 2 via node A as compared to 3 via node D. Node D selects ▲ over ■, since it learns this route directly from an external peer which is preferred in BGP. After Round 3, the BGP propagation process converges to a stable state and we can derive the forwarding plane as shown on the left in Figure 2. For completeness, we include the full BGP decision process in Appendix A.3.

**Configuration Parameters**   For our purposes, we define the set of synthesized configuration parameters as follows: For OSPF, we synthesize link weights as explored in existing work [15, 16]. For BGP, we focus on a setting, where we synthesize BGP import policies only. This means we synthesize the modifications required for BGP announcements when entering the network, to satisfy the routing specification. Previous work has confirmed that this is a realistic configuration setting that applies to a majority of real-world networks [9, 12, 38].

Based on the observation that routing protocols such as OSPF and BGP can be expressed as message-passing algorithms, we heavily rely on GNNs/NAR for the design of our synthesizer model as discussed next. In our evaluation, we then train such a model for the concrete case of BGP/OSPF and compare synthesis performance with a traditional SMT-based tool.

## 4   Neural Configuration Synthesis Model

We train a graph neural network (GNN) as the neural synthesis configuration model. The model's input consists of a topology, a specification, and a configuration sketch where to-be-synthesized parameters are omitted. Following the NAR paradigm, we first encode this synthesizer input in latent space. Then, we apply an iterative processor network based on the graph attention mechanism [42]. Last, we apply a decoder network to predict the values of omitted configuration parameters, thereby synthesizing a configuration. To remain protocol-agnostic, our model is generic with respect to the input format, using an intermediate representation based on Datalog-like facts. Figure 3 provides an overview of our graph-based neural synthesizer architecture.

### 4.1   Training Dataset of Inverse Pairs

A learning-based synthesizer model is formulated as a supervised learning problem. Thus, we can directly train a neural network to learn the inverse mapping, given a dataset of corresponding input-output pairs. To obtain such a dataset, we sample a random network configuration for some topology using a uniform generative process. Then, we simulate the involved protocols using $Prot$, to obtain the corresponding forwarding plane. Next, we extract a specification by randomly selecting properties that hold for the computed forwarding plane. This leaves us with a pair of specification and topology as input, and a corresponding configuration as output.

The key to constructing the dataset is the implementation of $Prot$. Even though $Prot$ is protocol-specific, it turns out the overall implementation effort is comparatively low, especially when compared to SMT-based synthesis methods. Protocols are well-defined algorithms that can be easily simulated, whereas the alternative of implementing hardcoded synthesis rules directly often requires expert

```
router(A)              network(N1)
router(B)              bgp_route(E,N1,2,3,1,0,1)
(...)
                       fwd(A,B,N1)
conn(A,B,2)            not fwd(B,A,N1)
conn(A,C,?)❶           (...)
(...)
```

Figure 4: A fact base encoding a network's topology, parameters such as link weights (`conn` facts) and a specification (`fwd` facts).

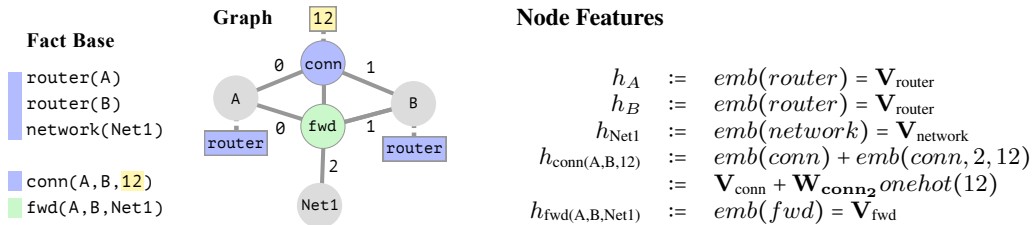

Figure 5: An example of embedding a simple fact base using our generic graph embedding scheme. Different neighborhoods are indicated as edge labels `0,1,2`. The full set of structural embedding rules is provided in Figure 6, Appendix A.1. We mark nodes and facts relating to topology, specification and configurations in color.

knowledge of SMT solvers. This process may also be adapted to rely on actual router hardware to compute the result of $Prot$, thereby capturing real-world behavior precisely.

### 4.2 Embedding Topologies, Specifications and Configurations

To remain agnostic with respect to routing protocols, our model architecture implements a generic graph-based encoding of Datalog-like facts, similar to knowledge graphs [33]: we first encode topologies, configurations and specifications as a set of Datalog-like facts and then employ a generic embedding scheme to embed these facts into latent space.

**Fact Base**   A set of Datalog-like facts, as depicted in Figure 4, serves as the input *fact base* $\mathcal{F}$ to our synthesizer model. $\mathcal{F}$ is a set of facts $f(a_0, \ldots, a_n)$ where arguments may be constants (e.g. A or B) or integer literals. Each fact has a corresponding boolean truth value denoted as $[f(a)]_\mathcal{B}$. For instance, from the fact base in Figure 4 we can derive $[fwd(A, B, N1)]_\mathcal{B} = \texttt{true}$ and $[fwd(B, A, N1)]_\mathcal{B} = \texttt{false}$.

**Synthesis as Completion Task**   A network's topology, protocol configuration, parameters, and the specification are all represented in a single fact base. To predict the value of unknown, to-be-synthesized parameters, we support the notion of unknown parameters as illustrated at ❶ in Figure 4, where the link weight between router nodes A and C is omitted. Based on this input format, synthesis corresponds to using our model to predict the value of unknown parameters in a provided fact base.

**Embedding**   To transform a fact base into a graph with node features, we apply a structurally-defined embedding scheme. An example of this embedding scheme is given in Figure 5. As shown, we embed topology, specification and configuration all into a single graph. This places network routers, the related specification predicates as well as configuration parameters in close adjacency to each other, simplifying the synthesis procedure for the processor network. In our scheme, both facts and constants are represented as distinct nodes. The relationships between facts and constants are encoded as node adjacency as defined by neighborhood functions $N_i$. We use multiple neighborhoods, i.e. multiple types of edges, to encode the argument position of a constant occurring in a fact. To handle unknown parameters, we replace the corresponding embedding with a learned $\mathbf{V_{hole}}$ embedding. Overall, the embedding function EMB relies on a set of learned parameters, including $\mathbf{V_f} \in \mathbb{R}^D$ per fact type in $\mathcal{F}$, $\mathbf{W_{bool}} \in \mathbb{R}^{D \times 2}$ for boolean values, $\mathbf{W_{f_i}} \in \mathbb{R}^{D \times N}$ per integer argument of fact type $f$ and $\mathbf{V_{hole}} \in \mathbb{R}^D$ to represent unknown parameters. $D$ is the dimensionality of the latent space and $N$

Table 1: Average consistency with $3 \times 16$ BGP/OSPF requirements, sampling configurations randomly from a uniform distribution and multi-shot sampling using our synthesizer model.

|        | Random          | 1-Shot              | 4-shot              | 8-shot              |
|-------:|-----------------|---------------------|---------------------|---------------------|
| Small  | 0.87±0.11       | 0.94±0.04           | **0.95**±0.03       | **0.95**±0.04       |
| Medium | 0.81±0.10       | **0.96**±0.04       | **0.96**±0.04       | **0.96**±0.04       |
| Large  | 0.80±0.05       | 0.93±0.05           | 0.93±0.05           | **0.94**±0.05       |

specifies the number of supported integer values. For the complete embedding scheme, please see Appendix A.1.

### 4.3 A NAR-based Synthesizer Model

Our synthesizer model employs an encode-process-decode architecture, inspired by neural algorithmic reasoning [41]. It employs four main components to produce the output distribution for an unknown parameter in a fact base $\mathcal{F}$: The fact base embedding $\text{EMB}$, the encoder $\text{ENC}_{GAT}$, the processor $\text{PROC}_{GAT}$ and the fact-type specific decoder network $\text{DEC}_{f_i}$.

Overall, the model can be expressed as follows, where $X_j$ and $H_j$ refer to the intermediate node representations per node $j$ in the graph constructed via our fact base embedding.

$$X_j := \text{ENC}_{GAT}(\text{EMB}(\mathcal{F}))_j + \mathbf{z} \quad \text{where} \quad \mathbf{z} \sim \mathcal{N}(0,1)$$
$$H_j := \text{PROC}_{GAT}(\{X_i\})_j \tag{2}$$
$$O_{f_j(a_0,...,a_{i-1},?,...,a_n)} := softmax(\text{DEC}_i^f(H_j))$$

$\text{ENC}_{GAT}$ is a GNN relying on Graph Attention Layers (GAT) [42] for propagation. We additionally apply noise to the latent node representation $X_j$ via $\mathbf{z}$ as a source of non-determinism, anticipating the fact that the synthesis problem often has more than one solution. The processor $\text{PROC}_{GAT}$ is modelled as an iterative process. It consists of a 6-layer graph attention module which we apply for a total of 4 iterations. According to the NAR paradigm, this computational structure encodes an inductive bias towards an iterative solution to the synthesis problem.

In the encoder and processor GNNs, we rely on a composed variant of the graph attention layer as introduced by [42]. We employ multiple graph attention layers in lockstep, one per neighborhood $N_i$ defined by our graph embedding (cf. Section 4.2), and combine the intermediate results after each step by summation. For details on the graph attention module, please refer to Appendix A.2.

Finally, an argument decoder $\text{DEC}_{f_i}$ is used, to produce the output distribution $O_{f_j(a_0,...,a_{i-1},?,...,a_n)}$ for an unknown parameter at position $i$ of fact $f_j \in \mathcal{F}$, corresponding to node $j$. For this, the synthesizer model provides one decoder per integer argument, per supported fact type. For example, a model for the fact base in Figure 4 would provide a decoder network $D_2^{\text{conn}} : \mathbb{R}^D \to \mathbb{R}^N$ to decode values for the third argument of a `conn` fact. Decoder networks are implemented as simple multi-layer perceptron models.

**Supervision Signal**  During training, we mask all fact arguments that represent to-be-synthesized configuration parameters in our synthesis setting as unknown parameters. As a supervision signal we use the masked values as the ground-truth and apply a negative log-likelihood loss to the output distribution of the corresponding argument decoder networks.

**Multi-Shot Sampling**  To sample values from the output distributions of unknown parameters, we apply a multi-shot sampling strategy. We sample values only for a subset of unknown parameters, insert them in the input fact base and run the synthesizer again. We repeat until no more unknown parameters remain. This multi-shot strategy allows the model to incorporate concrete values of previously synthesized parameters in its computation. In our evaluation, we compare this approach to sampling all parameters at once.

## 5 Evaluation

In this section, we assess the performance of our model trained for BGP/OSPF synthesis. For this, we trained a synthesizer model on a dataset of $10,240$ samples, constructed by randomly sampling

Table 2: Average specification consistency of our synthesizer model, where standard deviation is reported with respect to the different topologies in a dataset. We apply the model to synthesis tasks with $3 \times N$ requirements, i.e. $N$ requirements per supported specification fact.

| Dataset | | fwd | reachable | trafficIsolation | Overall | Full Matches | $> 90\%$ Matches |
|---|---|---|---|---|---|---|---|
| $3 \times 2$ | S | 0.97 | 0.94 | 1.00 | **0.96**±0.07 | 6/8 | 6/8 |
| | M | 0.95 | 0.94 | 1.00 | **0.94**±0.08 | 5/8 | 5/8 |
| | L | 0.92 | 1.00 | 1.00 | **0.94**±0.06 | 4/8 | 4/8 |
| $3 \times 8$ | S | 0.98 | 0.98 | 0.91 | **0.96**±0.05 | 4/8 | 7/8 |
| | M | 0.97 | 0.98 | 1.00 | **0.98**±0.03 | 4/8 | 8/8 |
| | L | 0.96 | 0.92 | 0.97 | **0.95**±0.03 | 1/8 | 8/8 |
| $3 \times 16$ | S | 0.98 | 0.92 | 0.95 | **0.95**±0.03 | 2/8 | 8/8 |
| | M | 0.95 | 0.95 | 0.98 | **0.96**±0.04 | 3/8 | 7/8 |
| | L | 0.94 | 0.91 | 0.95 | **0.93**±0.05 | 1/8 | 6/8 |

topologies, corresponding specifications and BGP/OSPF configurations as described in Section 4.1. For details on BGP/OSPF dataset generation and training, please see Appendix A.3. Lastly, we also evaluate our architectural design decisions in an ablation and parameter study in Appendix D.

**Dataset, Metrics and Experimental Setup**   We compare using datasets Small (S), Medium (M), and Large (L). Each dataset comprises 8 real-world topologies taken from the Topology Zoo [29], where the number of nodes lies between 0-18, 18-39, and 39-153, respectively. To obtain random forwarding specifications we use the same generative pipeline as discussed in Section 4.1. Regarding forwarding requirements, we implement support for three specification facts: `fwd` requirements to set/block forwarding paths, `reachable` to specify reachability and `trafficIsolation` to induce traffic isolation among traffic classes (no shared links). In each topology, we do synthesis for 4 different traffic classes (routing destinations) at a time. Overall, this results in 3 datasets x 8 topologies per dataset x 3 differently-size specifications = 72 synthesis tasks. To assess synthesis quality, we determine specification consistency as the relative number of specification facts in a fact base that are satisfied by the synthesized configuration. We run all experiments on an Intel(R) i9-9900X@3.5GHz machine with 64GB of system memory and an NVIDIA RTX 3080 GPU with 10GB of video memory.

## 5.1   Synthesis Quality

To assess the quality of synthesized network configurations, we examine specification consistency with increasingly large topologies and specifications. For each synthesis task, we run our synthesizer 5 times using 4-shot sampling and report the network configurations with the highest overall consistency. Table 2 documents the results. On average, our synthesis model achieves $> 93\%$ specification consistency for all datasets and specifications. With few requirements, the synthesizer model even succeeds in producing fully-consistent configurations (cf. Full Matches in Table 2). We observe a slight decrease in consistency with increasingly large topologies.

**Multi-Shot Sampling**   To determine the effect of multi-shot-sampling, we report consistency when using 1-shot, 4-shot and 8-shot sampling in Table 1. As a baseline, we also show consistency when sampling configurations from a uniform distribution (per parameter). This simple method can achieve suprisingly good results, as parts of a specification may be satisfied naturally by the mechanics of shortest-path routing. Still, competing with this baseline our synthesizer model shows clear improvements and multi-shot sampling further increases consistency.

**Number of Samples**   We also consider the number of times we sample from our synthesizer model. We observe that sampling more than one alternative configuration can lead to improved best-of specification consistency across all datasets. Based on this observation, we boost synthesis peformance by sampling multiple times per synthesis task. For each configuration that we obtain in this way, we simulate the routing protocols, obtain the forwarding plane and check specification consistency. This fully automated process allows us to determine the best result without consulting the user. We select the best result as the overall output for synthesis, dismissing the other samples. We experiment how the number of times we sample affects the resulting consistency in Appendix B.

Table 3: Comparing consistency and synthesis time of our method (Neural) with the SMT-based NetComplete. The notation n/8 TO indicates the number of timed out runs out of 8 (25+ minutes).

| # Requirements | | NetComplete (s) | Neural CPU (s) | Speedup | ∅ Consistency | Full Matches |
|---|---|---|---|---|---|---|
| 2 reqs. | S | $18.07s\pm14.55$ | $0.72s\pm0.54$ | **25.2x** | $0.97\pm0.09$ | 7/8 |
| | M | $60.86s\pm33.39$ | $3.18s\pm4.32$ | **19.1x** | $0.94\pm0.13$ | 6/8 |
| | L | $1389.48s\pm312.58$ 7/8 TO | $24.25s\pm28.35$ | **57.3x** | $0.99\pm0.03$ | 7/8 |
| 8 reqs. | S | $247.69s\pm436.90$ | $1.25s\pm1.02$ | **198.7x** | $0.96\pm0.08$ | 6/8 |
| | M | >25m 8/8 TO | $4.55s\pm4.30$ | **329.8x** | $0.97\pm0.04$ | 4/8 |
| | L | >25m 8/8 TO | $31.28s\pm28.53$ | **48.0x** | $0.97\pm0.05$ | 5/8 |
| 16 reqs. | S | $1416.83s\pm235.25$ 7/8 TO | $2.88s\pm1.66$ | **492.0x** | $0.92\pm0.06$ | 1/8 |
| | M | >25m 8/8 TO | $6.53s\pm5.10$ | **229.8x** | $0.95\pm0.05$ | 2/8 |
| | L | >25m 8/8 TO | $87.99s\pm141.97$ | **17.0x** | $0.95\pm0.03$ | 2/8 |

Overall, sampling more than once is beneficial for all datasets. Average best consistency values appear to be reached after 4-5 samples for $3 \times 16$ BGP/OSPF requirements. Hence, we rely on 5 samples in all other experiments as a trade-off of fast synthesis time and good specification consistency.

**Unsatifiable Specifications**   In practice, network operators may sometimes provide unsatisfiable specifications. While exact methods will typically return an error for such inputs, our relaxed setting enables us to consider partial solutions, i.e. configurations that still achieve high specification consistency while ignoring unsat requirements. This may be preferable in some scenarios, especially when perfect specification consistency is not critical anyway. To simulate this scenario, we evaluate specification consistency for OSPF-only synthesis tasks, which were all verified to be unsatisfiable using the SMT-based synthesizer NetComplete [15]. We still observe a comparatively high average consistency of 0.90 for our learned synthesizer. This suggests that our model is indeed capable of handling unsatisfiable specifications, while still producing good, partial solutions. For more detailed unsat results and methodology, see Appendix B.2.

## 5.2   Comparison to SMT-based Synthesis

We compare the synthesis time of our learned synthesizer with the SMT-based, state-of-art configuration synthesis tool NetComplete [15]. We compare BGP/OSPF and OSPF-only synthesis time. For each topology/specification, we report the total time of running our synthesizer 4 times using 4-shot sampling, whereas for consistency we report the best out of the 4 runs. If a fully consistent configuration is found, we do not continue to sample and only report time until then. For NetComplete, we time out at 25 minutes and report that as a lower bound if exceeded. We restrict our comparison to specifications that only include forwarding paths (i.e. `fwd` facts), as the type of supported requirements in NetComplete does not fully align with our model.

Table 3 shows that our synthesizer model outperforms NetComplete by multiple orders of magnitude. We observe a speedup of $20 - 490\times$ that increases with the size of topology/specification. The loss of precision due to approximation remains moderate, with the average consistency of the synthesized configurations being greater than $92\%$ even for large topologies. In comparison, NetComplete times out on more than half of the synthesis tasks, which means that we only observe a lower bound for speedup. Running our model on a GPU can result in even greater speedups (up to $900\times$ given enough GPU memory, cf. Appendix B.3). Comparing OSPF-only synthesis time, our synthesizer model achieves a speedup of 2-10x over NetComplete. With 16 requirements on dataset L this can also increase up to $500\times$. Further, our model produces fully-consistent OSPF configurations even more often than for BGP/OSPF. See Appendix B.4 for the full OSPF-only synthesis comparison.

## 5.3   Discussion

We have shown that our learning-based synthesizer reaches a very high degree of consistency, often producing fully-consistent configurations. With respect to synthesis time, we outperform SMT-based methods by a large margin, especially for larger topologies. However, this comes at a price: our model sometimes fails to produce fully consistent configurations, especially for large topologies

with many requirements. In comparison, SMT-based synthesis will always produce fully consistent configurations if and once it completes. We, therefore, observe a trade-off between consistency and synthesis time. We also note that our evaluation considers real topologies (Topology Zoo [29]) but not real specifications. This is due to the lack of a large, practical dataset thereof. Still, we experiment with robustness (Appendix D) by introducing distribution shift regarding the size of specifications during training and achieve comparable performance.

**Scaling to Even Larger Topologies**  The Topology Zoo [29] as used for our evaluation, provides a good range of realistically-sized networks. However, if we consider synthesis at very large scale (e.g. thousands of routers), we note the following scalability limitations: (1) Our models consume a lot of video memory, reaching beyond the amounts available on current consumer GPUs (> 12GB). This limit is reached with networks of 150 or more routers, and we have to do inference on the CPU (as indicated in Table 3). If even faster synthesis is important at this scale, more than one such GPU is needed for inference. Further, (2) the distance that information is propagated in the synthesizer GNN is finite due to the model's fixed number of iterations. This means that in very large networks, our synthesizer model will only be capable of deriving solutions by local reasoning which is very likely to impact synthesis quality. While out of the scope of this paper, longer training with more synthesizer iterations and larger topologies may be necessary to obtain comparable results at very large scale.

Nonetheless, we envision a wide range of practically-relevant applications for fast, approximate synthesis, including ML-guided synthesis, unsatisfiable specifications, and hybrid synthesizers, leveraging both learning and SMT solvers. We list a number of future directions in Appendix C.

## 6 Related Work

**Traditional Configuration Synthesis**  Next to methods based on compilation such as Propane/PropaneAT [5], there is a number of exact configuration synthesis methods based on constraint solving, e.g. ConfigAssure [31], SyNet [14] and NetComplete [15]. These tools are typically hand-coded, very protocol-specific, and can be slow due to the solvers they employ. In contrast, our approach is approximate but scales to much larger networks. Further, as a side-product of our learning-based approach, we can easily adapt to new protocols and allow for transparent cross-protocol reasoning, merely by training on different protocol data.

**GNNs and Networking**  DeepBGP [2] relies on GNNs and reinforcement learning to do configuration synthesis. However, it is limited to BGP configuration and is slower than SMT-based synthesis. In contrast, our learning-based framework is cross-protocol, does not rely on reinforcement learning and provides better synthesis times. Apart from configuration synthesis, GNNs have also been applied to other problems in the networking domain. For example, the authors of RouteNet [34] use GNNs to predict networking performance metrics. Other work focuses on learning improved protocols like Q-Routing [8] and Graph-Query Neural Networks [18].

**Neural Algorithmic Reasoning**  NAR [41] refers to the idea of replacing algorithms with neural networks to learn improved algorithmic procedures. Successful applications include graph algorithms [43], combinatorial optimization problems [10, 26, 37] and multi-task settings [23]. Our synthesis framework is the first application of NAR to the networking domain and relies on the NAR-native encode-process-decode architecture. In the hierarchy of NAR approaches in [10], our method is an algorithm-level approach, as we do not supervise on intermediate steps. Although step-level methods promise better generalization, it is not clear what an intermediate result of a general synthesis procedure would be. Future work on a step-level approach may further improve our model.

## 7 Conclusion

We presented a learning-based method to enable approximate but scalable network configuration synthesis. For BGP/OSPF routing, our neural synthesizer is up to 490× faster than SMT-based methods, while producing configurations with very high specification consistency. We believe there are future research that can be explored in the direction of learning-based synthesis and ML-guided network configuration. **Ethical Issues** This work does not raise any ethical issues.

**Acknowledgments** This work was partially supported by an ETH Research Grant ETH-03 19-2.

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
