# OpenReview forum: "Learning to Configure Computer Networks with Neural Algorithmic Reasoning"
_NeurIPS.cc/2022/Conference — NeurIPS 2022 Accept_

### Official Review · Reviewer_cd7P · 2022-07-12

**Rating:** 4
**Confidence:** 3
**Soundness:** 2 fair
**Presentation:** 2 fair
**Contribution:** 3 good

**Summary:**

This paper is concerned with the problem of finding a computer network configuration given a topology and a set of routing requirements.
Instead of relying on traditional Satisfiability Modulo Theory (SMT) approaches, the proposed approach builds upon recent advances in approximating computationally hard decision problems with Graph Neural Networks (GNNs). An advantage of the proposed technique lies in its protocol-agnostic nature: in principle, it can be used to configure any protocol that can be reduced to a message-passing computation over the network.
After training, the model is able to efficiently generate configurations that satisfy the requirements with high probability.

**Questions:**

1- What is the rationale for encoding everything (topology, configuration parameters and requirements) in one fact graph and embedding it into a single latent space?

2- Can your technique be applied to much larger topologies than those seen during the training?

3- How does your techique perform on topologies with thousands of nodes?

4- Have you considered different GNN architetures in your experiments?

5- Have you tried configuring protocols that are out of reach of the SMT-based configurators?

**Limitations:**

As mentioned above, I wish that the paper clarified how the proposed approach generalize after training to larger problems, since this is is crucial for applying the proposed technique to large, real-world networks.

**Strengths And Weaknesses:**

The paper is generally well-written and the proposed approach makes sense to me. While many papers aim at solving with GNNs more general decision or optimization problems, such as SAT or (weighted) model counting, I think it makes sense to introduce in the learner an inductive bias for solving specific real-world problems. I found interesting the realization that routing protocols can be cast as message-passing algorithms, thus their (approximate) configuration is amenable to GNN-based approaches.

I wish that some aspects were more formally defined and I struggled with understanding the rationale for embedding the different inputs into a single latent space.
Most importantly, I think that the experimental section should be improved in order to show the usefulness of the approach in real-world settings (see detailed comments below).

---

**Main comments**

---

I don't understand the advantage of mapping different inputs
(topology, specification and configuration sketch) into a single
embedding space instead of treating them differently. As the general
trend is aiming at finding semantically disentangled latent spaces,
mixing together in the same latent space facts related to the topology
and information on the protocol seems suboptimal. Can you elaborate on
this algorithmic choice?

---

"To remain agnostic with respect to routing protocols, our model
architecture implements a generic graph-based encoding of Datalog-like
facts"

Probably, adding a figure depicting the resulting graph for a small
fact-base would significantly help understanding the proposed
approach.

---

I wish that some archtectural choices were discussed in detail
and motivated.  For instance, what is the added benefit of having
GATs? Is adding a Gaussian noise to the input encoding a standard
practice when multiple solutions to the optimization problem exists?
Have the authors performed ablation tests for validating these choices?

---

On the experimental section:

- It is not clear whether the proposed model generalizes to larger
   topologies. This is is a pivotal aspect, unless the model can be
   effectively trained on real-world topologies with "millions of
   nodes".

- The benchmarks only include very small topologies. I get that the
   SMT-based approach doesn't scale to large topologies. Nonetheless
   Table 2 could show the average consistency of the generated
   configurations for much larger networks. After all, this is the
   whole point of approximating the problem with neural networks.

- Besides a comparison of the proposed approach with a SMT-based
   configurator, no alternative is considered. Why not reporting
   ablation studies for different architectural choices? These preliminary results are suggesting that GNN-based
   techinques can be used to approximate this complex optimization
   problem, not that the proposed approach and architectural choices
   are particularly well-suited for the task.

- The fact that the approach is protocol-agnostic seems one of its
   strongest selling points (see abstract), but in the experiments
   only a couple of protocols are considered. Plus, both protocols
   were already encoded in SMT. Are there protocols that are out of
   scope of (or currently not supported by) SMT-based methods? Adding
   additional experiments with more protocols would strenghten this
   aspect.

---

**Minor points**

---

In my opinion, the formal description of the problem in Section 2 could be improved:

- It is not initially clear what FWD is from a mathematical viewpoint. Figures 1 and 2 clarify that it can be represented as a graph, but I would mention it in the text anyway.

- It is initially mentioned that the topology T is fixed in the
  configuration synthesis task and it is thus omitted in the following
  equations, but it is reintroduced after line 98. In the same
  equation, Prot is not mentioned anymore. I get that the proposed
  approach to synthesis is protocol-agnostic, but that doesn't seem to
  be the case in general (e.g. with SMT-based approaches). It feels
  like the description of the task is mixed with specifics of the
  proposed approach.


---

- While I tend to like papers with many examples, I don't see the
purpose of 'Example intuition' at line 126. What is exactly the
intuition it should convey? Is it really functional to understanding
the contribution? On a side note, it also feels unnecessary for people
who know BGP and OSPF already, while being too vague to really help
understanding how these routing algorithms work.

---

"$D_2^{connected}∶ \mathbb{R}^D → \mathbb{R}^\mathbb{N}$"

- What is the subscript 2 indicating? I guess the superscript
$\mathbb{N}$ (typically referring to the set on natural numbers) is a
typo? Also, how is this output a distribution over configuration values?

---

"A learning-based synthesizer model is formulated as a simple
input-output mapping."

- I don't understand the sentence. I guess is that you cast the network
configuration given the topology as a supervised learning problem.

---

"Based on the observation that routing protocols such as OSPF and BGP
can be expressed as graph algorithms"

- I would write 'message-passing algorithms'.

---

> ### Author Response · Authors · 2022-08-02
> **Re: Review of Paper10848 by Reviewer cd7P**
>
> Thank you for your review, we appreciate and value your thorough feedback.
>
> > Q1
>
> This is an interesting point, especially your proposal on separating the latent spaces. To illustrate the rationale of our embedding, consider the newly-added Figure 5 in the updated revision: Our structural graph embedding places nodes representing network routers, related specification constraints and topological properties in close proximity to each other. The processor can then retrieve the information required for synthesis from the neighborhood of a parameter (e.g. a link weight). All available information is embedded into a joint latent space in which we learn the actual synthesis procedure. This mapping to high-dimensional space allows the processor maximum flexibility to combine and compress topological, configuration and specification information in a suitable form, without being bottlenecked by a manually selected format or separation. Previous work on NAR [40] compares this to the way humans apply classical algorithms to noisy, raw data, by (heuristically) deciding how to compress raw data complexity into inputs the algorithm will accept. Lastly, by combining all available information into a single latent space, our system remains agnostic w.r.t. the supported types of topologies, protocols and specifications, as no manual modeling is required when introducing new types of links or protocol features.
>
> Nonetheless, we were interested in experimenting with separating the latent spaces and included a corresponding experiment in our ablation study (see newly-added Appendix D.1 for details). Due to the increased computational requirements of these models we could not fully finish training, but report results up to epoch 2000 (instead of 2800). Overall, we did not observe a performance benefit of separating the latent spaces using multiple encoder+processor GNNs.
>
> > Q2
>
> Yes, our technique indeed scales to much larger networks than those seen during training and the presented evaluation already demonstrates this capability. To illustrate, we added a visualization and discussion of the distribution shift when it comes to the size of networks and specifications D.2 (newly-added Appendix D). Our dataset statistics show that our generated training dataset contains only comparatively small networks (15-25 nodes) and specification, whereas the networks of the Topology Zoo [26] (4-153 nodes) and specifications during evaluation are often much larger.
>
> > Q3
>
> Thank you for raising this. We agree that scalability to networks of realistic size is central to our work. For this, consider that our synthesis setting (BGP/OSPF) relates to configuring backbone networks as operated by e.g. internet service providers (ISPs). As a reference for the size of these networks, we rely on the Topology Zoo [26], a widely-cited benchmark collection of topologies, commonly used to evaluate tools in the networking community when it comes to showing real-world applicability. The Topology Zoo comprises networks with up to 200 nodes, whereas the largest network we use in our evaluation has 153 nodes. The larger part of the samples in this dataset however, is of much smaller scale. Overall, we therefore claim that our evaluation already considers realistically-sized networks.
>
> Still, we note the following limitations of our models when scaling to thousands/millions of nodes:
>
> * Memory requirements: Our models already consume a lot of memory (>12G) for ~150 nodes and we have to do inference on the CPU.
>
> * Even though our synthesizer models are iterative and employ multiple GNN layers, the maximum distance that information can travel by propagation is finite. This means that in very large networks, our synthesizer model will only be capable of deriving solutions by local reasoning, which is very likely to impact synthesis quality.
>
> > ablation and architectural design choices including different GNN architectures (Q4)
>
> We included our ablation and parameter study in the new D Additional Experiments part of the updated appendix. Among other things, this includes experiments with different GNN architectures, employed internally by our synthesizer models, as well as a demonstration of the effectiveness of adding gaussian noise, especially as we sample multiple solutions from a synthesizer model.
>
> > Q5
>
> For the work at hand, we focused our efforts on a thorough evaluation of BGP/OSPF synthesis as well as OSPF-only synthesis. However, we plan to build upon this work and apply our largely protocol-agnostic system to other settings. Most protocols can be modeled by SMT-based synthesis, but performance varies. Further, we see great potential in applying our learning-based system to specifications which include “soft” constraints that can not easily be interpreted by symbolic systems like SMT. You can also find an in-depth discussion of future directions in Appendix C.
>
> Thank you also for your minor notes, we agree with your remarks and have revised accordingly.

---

> > ### Comment · Reviewer_cd7P · 2022-08-07
> > **Response to the authors**
> >
> > Thank you for your detailed answer. Overall, I think that the presentation significantly improved in the last version of the manuscript. I also appreciated the additional ablation studies and the in-depth discussion of future directions included in the appendix.
> >
> > > Still, we note the following limitations of our models when scaling to thousands/millions of nodes: [...]
> >
> > I think that it would be helpful to make these limitations clear in the main text.

---

> > > ### Author Response · Authors · 2022-08-08
> > > **Re: Response to the authors**
> > >
> > > Thanks, yes. We included a note on scaling in the discussion of our evaluation.
> > >
> > > Edit: We have now uploaded a revised PDF including this change.

---

### Official Review · Reviewer_CE15 · 2022-07-20

**Rating:** 8
**Confidence:** 3
**Soundness:** 3 good
**Presentation:** 3 good
**Contribution:** 4 excellent

**Summary:**

The authors put forward a novel method to automatically reconfigure computer networks. Their main contribution is to relax the notion of an acceptable configuration to account for the cases where only an approximation is possible. The latter case is chosen so that various learning techniques can be applied.

The main contributions of this work are: A relaxation of the configuration synthesis problem, that admits a scalable approximate solution, and the introduction and training of a neural network that generates configurations likely to satisfy the approximate guarantees mentioned above, in a *scalable* fashion (this latter point is what makes this work interesting).

**Questions:**

Can you please make Section 1 a bit more reader-friendly? Thanks!

Throughout the paper there is a lot of undefined technical jargon (OSPF, BGP, etc) :) Could you please amend this?

Could you please explain Figure 1 a bit further?

**Limitations:**

In my opinion, the authors have adequately addressed the limitations and potential negative societal impact of their work.

**Strengths And Weaknesses:**

Originality:
This paper seems original. Although there exists some prior work on automating network configuration by using a variety of synthesis techniques, this work seems to be the first to present a trade-off between the accuracy of the chosen configuration and the speed of configuration. Moreover, this work seems to be the first in using learning-based techniques in the context of automating network configuration (in contrast to previous work, where SMT techniques are predominantly used).

Quality:
The quality of the paper is high, in my opinion: It is well written and the results seem strong.

Clarity:
Presentation of the paper and writing are clear, in general. I believe that Section 1 could have been written in a more reader-friendly way, especially to account for the case where the reader is not an expert and may not be familiar with SMT solvers, etc. (Could you please make this part a bit gentler?)

Significance:
This work is significant in my opinion; see Originality above.
Another reason I find this paper important is the scalability of the proposed neural network solution.

---

> ### Author Response · Authors · 2022-08-02
> **Re: Official Review of Paper10848 by Reviewer CE15**
>
> Thank you for your review, we appreciate and value your feedback. In addition to our response below, please also consider our revised PDF where we highlighted the key changes we made, especially regarding your notes on Section 1-3.
>
> > Can you please make Section 1 a bit more reader-friendly?
> > Throughout the paper there is a lot of undefined technical jargon (OSPF, BGP, etc) :) Could you please amend this?
>
> Thank you for bringing this up, your perspective is very important to us as we already came into this work with a networking and SMT background. We are actively working on further improving our draft to make sure our results are more accessible to a broad ML audience. In our revised draft, we have overhauled Section 1-3, to include more details on SMT, networking fundamentals and NAR where applicable. We also largely refrained from mentioning networking jargon until defined, wherever possible. We are still looking for ways to improve this but are also bound by space constraints.
>
> > Could you please explain Figure 1 a bit further?
>
> Figure 1 illustrates the process of network configuration synthesis. We focus on the upper level first: Computer networks employ a number of routing protocols to enable resilient and policy-adhering routing behavior. Operators provide a configuration $W$, composed of a set of parameters (e.g. link weights, BGP import policies). After deploying the configuration, the routing protocols are executed on the routers of a network, running as a distributed process. As the result of this computation each router derives its _forwarding table_, a mapping of incoming traffic to one of its neighbors. In sum, these forwarding tables form the so-called _forwarding plane_ Fwd which can be represented as a directed graph that specifies how traffic flows through the network. No matter the configuration, most routing protocols are designed to find at least some sort of forwarding plane (e.g. shortest path routing). However, in practice network operators often have requirements regarding the concrete forwarding behavior of their network. For instance, they may want to enforce or prohibit certain paths for performance and load-balancing purposes. This is where synthesis comes in: As shown on the lower level of Figure 1, synthesis automates the process of finding a configuration $W$ that produces a forwarding plane which satisfies some given specification $S$. In that sense, synthesis can be understood as the inverse function of executing the routing protocols and checking consistency with a specification, a process that otherwise requires a lot of manual network engineering and knowledge about the semantics of the involved routing protocols.
>
> In our revised draft, we included more discussion of Figure 1 in both Section 1 and 2 to better illustrate the problem to readers.

---

### Official Review · Reviewer_5wwX · 2022-07-21

**Rating:** 6
**Confidence:** 2
**Soundness:** 2 fair
**Presentation:** 2 fair
**Contribution:** 3 good

**Summary:**

This paper presents a GNN-based method for synthesising network configurations for a given network topology, such that most, if not all, specifications are satisfied.

The authors argue that existing, SMT-based, methods are of limited use, because they are solving an NP-hard problem exactly, and are thus inherently unable to scale to large instances. Since modern-day applications require fast network configuration generation for large networks, the authors propose to instead use an end-to-end, learning-based method to synthesise "good" network configurations, given a network topology and a set of requirements. Here, they relax the notion that all requirements must be met, and instead just aim to have as many of them met as possible. If I understand correctly, the idea is that this can be given to a human who can then adjust the configuration in order to meet all requirements. The authors argue that this is better than a solver that just doesn't terminate, or takes too long.

Futhermore, they argue that their method is protocol-agnostic, in contrast to hand-coded SMT methods, and realise this by encoding protocols using `Datalog`. In addition, they claim that their method can be used for cross-protocol reasoning. They limit their configuration synthesis to generating link weights for the OSPF protocol, and import policies for the BGP protocol, arguing based on the literature that this is a realistic choice.

To evaluate the quality of their synthesised configurations, the authors define the notion of 'specification consistency', which is simply the fraction of the requirements that are met. Their methods aim to find a protocol (e.g., link weights) that maximises this specification consistency.

The authors describe how they generated a training set of $10\:240$ (input, output) pairs, where the input is a topology and a specification of requirements, and the output is a configuration, such that they could train their models.

They evaluate their method primarily on solution quality, reporting averaged specification consistencies, with respect to three forwarding requirements. They do this for network topologies in three size categories, containing up to 153 nodes. They evaluate different sampling strategies for completing partial configurations into full ones. Finally, they evaluate the time needed to get their configurations, comparing that to the time required by an SMT-based exact method: `NetComplete`.

**Questions:**

Q1: I am confused by the paragraph in lines 85-91. The way that the configuration synthesis problem is described in lines 21-22 gives me the impression that it's a satisfiability problem: we need a configuration that meets the requirements. Then, in the paragraph in lines 75-84, the authors formulate an optimisation version of that satisfiability problem, which makes sense to me in the context of this work. However, in lines 85-86, they "propose to relax both the optimality as well as the rigid satisfaction requirements", which implies that the original configuration synthesis problem is an optimisation problem. Since understanding the exact problem that you are trying to solve is quite crucial, it would be great if the authors could clear this up for me and explain what they mean by the sentence in lines 85-86?

Q2: What exactly do the authors mean by "cross-protocol reasoning", and do they have any theoretical or empirical evidence that their method does indeed enable cross-protocol reasoning? If yes, (how) did they present it in the paper? If they did not present it in the paper, why not? If they do not have the evidence, can they please clarify what justifies this claim?

Q3: The authors mention computational complexity as a limiting factor for any method that aims to synthesise network configurations. Can they specify what that complexity is?

Q4: If I understand correctly, the authors have 3x8 = 24 instances, and they do synthesis for 4 different traffic classes each, and then with 2, 8 and 16 requirements per specification fact? So they have a total of 3x8x4x3 = 288 synthesis tasks. Is that correct? As I understand it, the authors then run the synthesiser 5 times, and report the configuration with the highest overall consistency, which I guess means averages over the `fwd`, `reachable` and `trafficIsolation` forwarding requirements? Am I correct in my understanding that for each of these 96 synthesis tasks, the authors did 5 runs and reported on the quality of the best run? If no, could the authors clarify this matter for me? If yes, could the authors report on the qualities of the other runs?

Q5: It seems that the authors assign the same weight to all the requirements. Since they propose essentially an optimisation method, for which we can measure the quality of the solution, I am wondering if the authors considered how realistic this is? They seem to expect a human to finalise the configuration to something that satisfies all requirements. I can imagine that some requirements are easier for a human to meet than other, so maybe some requirements could/should have less weight than others? I missed any reflection on this. Maybe the authors can clarify?

Q6: The choice to run the algorithms for 25 minutes seems strange to me. I could understand 15 minutes, I could understand 30 minutes, but 25 minutes smells a bit like cherry picking to me. Can the authors motivate this choice?

Q7: This may be just a gap in my knowledge, but what is a "strong inductive bias"? The authors mention it a few times and it seems important, but it is not clear to me what it is.

Q8: Can the authors explain why it is so important that their method does not rely on reinforcement learning?

Q9: Line 272-273: "We compare both BGP/OSPF and OSPF-only synthesis time. My first question is: why? My secon question: I don't see that reflected in the table, how should I read the table to understand it?

Some smaller remarks:

- line 15: 'resp.' Why 'resp.'? It doesn't make sense to me in the sentence structure.
- line 18: "Figure 1 - here" -> "Figure 1. Here"
- line 31: "inherent computational complexity" I find this statement rather unconvincing without specifying what that computational complexity is.
- line 36: "relaxing" -> "to relax"
- line 36-37: "to admit approximate solution with high utility -- configuration which may not always satisfy all requirements" I cannot parse this sentence, could you please rephrase?
- line 71: is the second $i$ subscript a typo?
- line 130: "the same destination" which is ...?
- line 134: "the decision cannot eliminate" I'm confused at the concept of a decision making decisions.
- line 138: "the left" do you mean "the right"?

**Limitations:**

The authors did not discuss any potential negative societal impact. I don't think that their work would have any, so I wouldn't hold this choice against them.

As for other limitations: the authors describe how they modified the original problem and proposed a method for solving that modified

**Strengths And Weaknesses:**

### Strengths

- As far as I can tell, the method that the authors propose in this paper is indeed a novel way of synthesising network configurations.
- The authors seems to have a decent grasp of how their work is positioned in the literature.
- The many paragraph titles in the text make it relatively easy to find relevant information in the paper.
- The text does not contain many grammatical errors.
- It seems to me that the problem and the proposed solution would be of interest to the NeurIPS audience.
- The experimental evaluation makes sense to me, although I have a couple of comments (see below).
- The results look promising.
- I like that the author explored the performance of their algorithm on unsatisfiable requirements also, as I can imagine that they might occur quite often in real-life, and their method has the benefit of giving the poor human who has to fix it at least some hints at how to do it.
- The authors to some extent explored different hyperparameter settings for their method (in Table 1).

### Weaknesses

- In general, I find that the authors are overselling their work. They use the phrase "radically new method/approach", without really showing what is so "radical" about their approach, but I'm willing to see this as a choice of words used for advertising that I personally find inappropriate, and hence a matter of personal taste. However, there are some more serious cases of possible overselling:
  - The authors state in the abstract that their approach "enables cross-protocol reasoning". This sounds very interesting, so I was excited to read the experimental section to get a sense of how well this works. Unfortunately, I could not find any theoretical or empirical evidence to back up this claim. Maybe they showed it, but it just wasn't clear to me what exactly they mean by "cross-protocol reasoning" (I am not very familiar with computer network configuration)? In that case, the paper would be improved (at least for me) by making this more clear.
  - I find the phrasing "breaking the scalability barrier" also problematic. The authors talk about the "inherent computational complexity" of the synthesis problem, without specifying what that complexity is, except for mentioning that parts of the problem are NP-hard. In addition, they modify the original configuration synthesis problem from a satisfiability problem to an approximation/optimisation problem. Nothing inherently wrong with that, provided that this is well-motivated, but I find that "breaking the scalability barrier" conveys a larger promise than "proposing a method to solve a different problem than the original one".
  - The authors claim in the abstract that the configurations that they synthesise "on average satisfy more than 93% of the provided requirements". However, the way I understand the sentence in lines 244-245, the authors obtain this number by always reporting the best out of five runs of each synthesis task. Maybe I misunderstand their experimental setup, but as I understand it now, it seems somewhat misleading to me. As far as I can tell, the authors do not report in the paper about the quality of the 4 discarded solutions per synthesis task. In a different experiment (Table 3) they seem to pick the best out of 4 runs instead of 5, which seems arbitrary and smells of possible cherry picking to me.
  - The observed speedup of upt to 560x reported in line 280 is somewhat undercut by the fact that the proposed method only has a full match in 2/8 instances for that case. I don't think that 25 minutes is a truely prohibiting time for an SMT-solver to come up with an exact solution, so while this speedup is impressive, I would argue that the goal of speedup is not autotelic, but only makes sense if you are also achieving similar quality results, and/or the original time was prohibitively long.
- The authors argue that the fact that routing protocols can be expressed as graph algorithms motivates their choice to "heavily rely on GNNs/NAR". However, they don't explain very well what GNNs are, and only explain NAR in the last paragraph of the Related Work section, right before the conclusion. It would've been easier for me to understand the points made in this paper if this concept was explained much earlier.
- I honestly have trouble verifying the validity of the experiments, because the experimental protocol is rather opaque to me. I have tried to add some clarifying questions below.

I address the other weaknesses in my questions below.

### Summary

My general impression of the paper is that it is a good attempt at solving a somewhat artificial problem. While fairly well-written, I found it hard to follow, but that is probably due to my lack of knownledge in the are. The experiments seem sensible, but the experimental setup is somewhat opaque to me. My largest issue is with the way in which the findings are presented, which I find somewhat misleading in some cases. Perhaps I am misunderstanding the claims or the methods in this paper, but in that case, it would probably benefit from some editing.

---

> ### Author Response · Authors · 2022-08-02
> **Re: Review of Paper10848 by Reviewer 5wwX**
>
>
> > Q1
>
> Traditionally, the original configuration synthesis problem, as solved by e.g. SMT-based synthesis, is not an optimisation problem. However, for this work we framed it as such for two reasons: (1) We think the framing helps illustrate the idea of synthesis to an ML audience, which may not necessarily be familiar with SMT solvers. And, (2) framing as an optimization problem allows us to consider solutions (network configurations) which are not globally optimal, but still provide high utility. This kind of relaxation is quite common in ML but not with SMT.
>
> > Q2: “What exactly do the authors mean by "cross-protocol reasoning?".
>
> By cross-protocol reasoning we refer to the ability to reason about multiple interacting protocols during synthesis (e.g. BGP depends on OSPF). Our method considers multiple interacting protocols as one system for which the synthesis procedure is learned end-to-end. This naturally enables cross-protocol reasoning. Our evaluation shows the effectiveness of our models at cross-protocol BGP+OSPF synthesis. Comparable SMT-based tools (e.g. NetComplete) implement multiple synthesis procedures (one for BGP and one for OSPF + logic to communicate between the two), requiring lots of engineering effort and extensive domain knowledge. In the updated revision, we further clarify this terminology in section 3.
>
> > Q3
>
> The computational complexity of configuration synthesis generally depends on the protocols involved. For our example setting of BGP/OSPF synthesis, we note that the underlying inverse shortest path problem of OSPF synthesis has been shown to be NP-hard [7]. Additional complexity results can be found in [16,43].
>
> > Q4 “they have a total of 3x8x4x3 = 288 synthesis tasks. Is that correct?” Am I correct in my understanding that for each of these 96 synthesis tasks, the authors did 5 runs and reported on the quality of the best run? If no, could the authors clarify this matter for me? If yes, could the authors report on the qualities of the other runs?
>
> Synthesis must be done for all 4 traffic classes in one run (OSPF link weights are class-independent + constraints like trafficIsolation span multiple classes). Overall, this means there are 3 datasets x 8 topologies per dataset x 3 differently-size specifications = 72 synthesis tasks. We have added a clarifying note to the related paragraph.
>
> Regarding the quality of the different runs per synthesis task, see our discussion above: We choose the best run automatically, meaning the quality of the other runs will be lower or the same as the final result.
>
> > Q5 Since they propose essentially an optimisation method, for which we can measure the quality of the solution, I am wondering if the authors considered how realistic this is?
>
> This is a good observation and an interesting direction for future work. Indeed, users are likely to have some sort of priority per constraint they specify, especially in a relaxed setting. Our current models do not support weighting the specification, nor does our metric of consistency. In Table 2, we do report specification consistency per type constraint, which already gives some relevant insight, but we will add a note about this possible extension of the general problem of approximate configuration synthesis.
>
> > Q6
>
> This choice is quite technically motivated: Our evaluation scripts expect timeouts to be specified in seconds (25 minutes = 1500s). The value was reached by experimentation, and increased to a level such that our finite computational abilities were not exceeded.
>
> > Q7
>
> This term describes the model's architecture which implements an inductive bias for learning a certain reasoning process for synthesis (in our case: graph-based representation, different edge types, iterative decoder). The idea of guiding the reasoning process in this way is inspired by NAR [40].
>
> > Q8
>
> NAR [40] specifically describes the concept of learning algorithmic procedures with strong supervision. RL offers an alternative to this by relying on weaker supervision but can be less sample efficient. We implement NAR, so we emphasize the difference to RL.
>
> > Q9
>
> We evaluate OSPF-only synthesis because it is an interesting problem on its own with its own line of preceding work (cf. [7, 16]). Due to space constraints, OSPF results are only given in text (line 285) with the full table in Appendix B.4.
>
> > negative societal impact
>
> We have added a note clarifying that our work does not raise any ethical issues.
> Thank you also for your smaller remarks, we will revise our draft accordingly.

---

> ### Author Response · Authors · 2022-08-02
> **Re: Review of Paper10848 by Reviewer 5wwX**
>
> Thank you for your feedback, we value and appreciate your review. Below we respond to your general concerns and in another comment to your questions. Please also consider the revised PDF with highlighted changes.
>
> > "overselling"
>
> Thank you for raising this, we have no intentions of overselling the significance of our results. In our work-in-progress, revised draft we put special focus on this matter and tried our very best to tone down our language where it seemed overly exaggerated.
>
> > “I find that "breaking the scalability barrier" conveys a larger promise than "proposing a method to solve a different problem than the original one".
>
> Thanks for pointing this out, indeed the hardness of the exact synthesis problem remains “unbroken”. Instead, our claim is directed at the practical level and the use of synthesis for network configuration in general. As described, SMT-based tools are very powerful but have scalability issues, which leads to extremely long synthesis times (sometimes >24h for a single run). So far, this barrier has prevented the wide-spread use of these tools in practice. The proposed method relaxes the problem and provides a less precise but faster and more practical method. We consider this an exciting development and therefore chose this wording. However, we appreciate your rigor and have revised our draft, completely refraining from using “breaking” terminology.
>
> > “The authors claim in the abstract that the configurations that they synthesise "on average satisfy more than 93% of the provided requirements". [...] However, the way I understand the sentence in lines 244-245, the authors obtain this number by always reporting the best out of five runs of each synthesis task.”
>
> This is likely a lack of clarity in our draft: We sample from our synthesizer model multiple times, producing a new configuration on each run. We then simulate the routing protocols for the produced configuration, obtaining the forwarding plane which in turn allows us to check specification consistency. This is a fully automated process and automatically determines the best result without consulting the user. We select the best result as the overall output for synthesis, dismissing the other samples. We added a more explicit discussion of this practice in the revised draft (l.284).
>
> The “Number Of Samples” experiment in our evaluation demonstrates that more samples tend to produce better results. For this reason we chose a lower number of samples when comparing synthesis times. Still, we share your concern about this and ran further experiments, additionally reporting both consistency with only 4 samples and synthesis time with 5 samples (cf. App. D, Additional Experiments). For both the synthesis time and quality, we can produce very similar results to those shown in the original paper. Also, we have revised our draft such that all main results are now obtained by sampling 5 times, favoring quality over synthesis time.
>
> > “The observed speedup of up to 560x reported in line 280 is somewhat undercut by the fact that the proposed method only has a full match in 2/8 instances for that case. I don't think that 25 minutes is a truly prohibiting time for an SMT-solver to come up with an exact solution”
>
> We agree that indicated speedup cannot be interpreted as pure speedup as the result is not the same. We try to emphasize this by always mentioning the entailed loss in precision (i.e. <1.0 consistency). On the other hand, still having occasional full matches, actually means that sometimes our method is able to compete as exact synthesis tools too.  Still, for use cases where exact (full match) solutions are absolutely necessary, our method is not applicable (cf. evaluation discussion). We try to clarify this from section 2 on, given that we do not aim to solve the exact synthesis problem to begin with. We have and will revise our draft further to explicitly state this wherever applicable.
>
> Regarding SMT synthesis time, we agree that 25m may not appear truly prohibiting, but note that this timeout is also caused by the limits of our computational abilities. Previous works [14,15] observe synthesis times of > 24h for networks similar in size (64 nodes) to our evaluation networks. In contrast, our method is less precise but faster (<90s even for 153 nodes). Theoretically this enables almost interactive usage, where results can still be tweaked by users, a common practice also with the result of SMT tools.
>
> > “The authors argue that the fact that routing protocols can be expressed as graph algorithms motivates their choice to "heavily rely on GNNs/NAR". [...] they don't explain very well what GNNs are, and only explain NAR in the last paragraph of the Related Work section, right before the conclusion. “]
>
> Thank you for this comment. In our work-in-progress, revised draft we have already extended the discussion of NAR in Section 3. We are still looking for ways to add more detail, but are also limited by space constraints.

---

### Official Review · Reviewer_yuPQ · 2022-07-25

**Rating:** 6
**Confidence:** 3
**Soundness:** 3 good
**Presentation:** 2 fair
**Contribution:** 3 good

**Summary:**

This paper studies the problem of finding a network configuration, such that, after applying a routing protocol, the resulting forwarding plane satisfies the given specifications. The proposed method involves encoding the specifications as well as an incomplete configuration (with missing parameters) as a fact graph (with multiple neighborhood functions) and then using a graph neural network model to predict the missing parameters (to complete the configuration). A synthetic dataset is generated for training the model (by first generating the configurations and then a set of specifications satisfying the resulting forwarding planes). This method has been demonstrated to be significantly faster than SMT-based methods (up to 500x faster) and on average satisfies around 93% of the given requirements.

**Questions:**

+ It appears that generating configuration parameters uniformly at random already achieves around 80% satisfaction rate. Is there an explanation for why this is the case?

**Limitations:**

Limitations regarding scalability are discussed in the paper.

**Strengths And Weaknesses:**

### Strengths

+ __Graph encoding enables protocol agnostic application.__ The overall technique can be applied w.r.t. any routing protocol since the fact graph encoding used is very general and furthermore, the data generation process only requires the ability to simulate the protocol.
+ __Simplicity.__ The proposed method appears simple and elegant.
+ __Can obtain approximate solutions even for unsatisfiable specifications.__ This aspect is interesting and I believe that it is of practical significance.

### Weaknesses

- __Lack of clarity in writing.__ Although the overall approach seems simple, the paper was a bit difficult to follow requiring multiple passes to understand.
    + Many network-related concepts are not defined, making the paper less accessible to the broader NeurIPS audience. For example, a forwarding plane is mentioned as the output of a protocol without defining what it actually is, mathematically.
    + The graph encoding seems to be one of the primary components of the technique and it is not described well in the main text. A diagram of the graph with different types of edges for the running example will be helpful.
- __Evaluation on synthetic specifications only.__ The primary evaluation is performed on randomly generated specifications. However, this might not capture the distribution of specifications encountered in realistic scenarios.

---

> ### Author Response · Authors · 2022-08-02
> **Re: Official Review of Paper10848 by Reviewer yuPQ**
>
> Thank you for your review, we very much value and appreciate your feedback. Below we respond to your concerns and questions.
>
> > “Many network-related concepts are not defined, making the paper less accessible to the broader NeurIPS audience.”
>
> Thank you for bringing this up, your perspective is very important to us as we already came into this work with a networking background.  We are actively working on improving our draft and have already uploaded a work-in-progress, revised draft of our paper in which we discuss networking fundamentals and the forwarding plane specifically in more detail. Our overarching goal is to make our results accessible to a broad ML audience.
>
> > “The graph encoding [...] is not described well in the main text.”
>
> This is an unfortunate consequence of space constraints and we had to include ~1 page of additional information and all structural embedding rules in appendix A.1 (see supplementary material). In our revised draft, we have extended on our discussion in the main text, adding Figure 5 which illustrates the encoding with a concrete example.
>
> > “Evaluation on synthetic specifications only.”
>
> This is a legitimate concern, so thank you for raising this. We acknowledge that transferring our results to real-world network configuration may entail a specification distribution shift. To address this, we carried out additional experiments in the newly-added appendix D.2 (revised PDF) and compiled dataset statistics (cf. Fig. 10). On one hand, we already observe a distribution shift when it comes to the size and shape of networks as well as the size of the specification, comparing our synthetic training dataset and our evaluation datasets (real topologies + synthetic specifications). This suggests that our synthesizer model exhibits some level of robustness regarding distribution shift. We further trained a synthesizer model on a newly-constructed training dataset with much smaller specifications and achieved similar synthesis performance (cf. Table. 6) which supports this hypothesis. We also added some discussion of this aspect to our evaluation in the revised draft. Overall, given that our work demonstrates that learning-based synthesis is viable (for real topologies, but synthetic specifications), we believe that future work may further build upon ours and incorporate real-world specification and even real-world hardware behavior (to evaluate but also during training).
>
> > “Question: It appears that generating configuration parameters uniformly at random already achieves around 80% satisfaction rate. Is there an explanation for why this is the case?”
>
> According to our observations this is caused by the structure of the considered network topologies. For instance, some topologies in the Topology Zoo (and the real world) contain constructs such as islands and star-like arrangement of routers, caused for instance by geographic or infrastructure constraints. Now, since our generative process for specifications only generates satisfiable synthesis tasks, some parts of the required specifications may be satisfied naturally by the mechanics of shortest-path routing, no matter the configuration. For example, consider a network where  some island of routers is only accessible via a specific link. If we assume shortest-path routing and specify that traffic must flow via this link, any set of chosen link weights will trivially satisfy this. Therefore, we include random sampling as a baseline and demonstrate that the synthesis capabilities of our models are clearly superior to simple methods such as random sampling.

---

> > ### Comment · Reviewer_yuPQ · 2022-08-08
> > **Re: Re: Official Review of Paper10848 by Reviewer yuPQ**
> >
> > Thanks for the rebuttal. I appreciate the changes to the writing making it more understandable.

---

> > > ### Author Response · Authors · 2022-08-08
> > > **Re: Re: Official Review of Paper10848**
> > >
> > > Thanks for your response.
> > >
> > > A small extension on our point on evaluating only on synthetic specification:
> > >
> > > Apart from our additional experiments, please consider that to our knowledge, no datasets of real-world specifications and corresponding topologies exist to date. This makes such an evaluation difficult. Given this situation, we tried to address this point by experimenting with the robustness of our model, when the distribution of specifications changes. We mention this in the revised draft already (evaluation discussion) but accidentally did not include this in our response above. Please apologize the oversight.

---

### Meta-Review · Area_Chair_Vj7j · 2022-08-24

**Recommendation:** Accept
**Confidence:** Certain

**Metareview:**

The reviewers found the approach proposed in the work to be novel and reviewers find the work to have promise. The reviewers also felt that it would be best for authors to present a detailed analysis on the scalability limitations and we are confident that authors would do so in the final version.

**Award:**

No

---

### Decision · Program_Chairs · 2022-09-14

Accept